# Bacterial Endophytes: The Hidden Actor in Plant Immune Responses against Biotic Stress

**DOI:** 10.3390/plants10051012

**Published:** 2021-05-19

**Authors:** Nadira Oukala, Kamel Aissat, Victoria Pastor

**Affiliations:** 1Laboratory of Ecological Microbiology, Department of Microbiology, Faculty of Nature and Life Sciences, University of Bejaia, 06000 Bejaia, Algeria; nadira.oukala@univ-bejaia.dz; 2Department of Microbiology and Biochemistry, Faculty of Nature and Life Sciences, University of Batna 2, 05000 Batna, Algeria; 3Metabolic Integration and Cell Signaling Laboratory, Plant Physiology Section, Departamento Ciencias Agrarias y del Medio Natural, Universitat Jaume I, 12071 Castelló de la Plana, Spain

**Keywords:** priming, endophytic bacteria, ISR, pathogens, signaling

## Abstract

Bacterial endophytes constitute an essential part of the plant microbiome and are described to promote plant health by different mechanisms. The close interaction with the host leads to important changes in the physiology of the plant. Although beneficial bacteria use the same entrance strategies as bacterial pathogens to colonize and enter the inner plant tissues, the host develops strategies to select and allow the entrance to specific genera of bacteria. In addition, endophytes may modify their own genome to adapt or avoid the defense machinery of the host. The present review gives an overview about bacterial endophytes inhabiting the phytosphere, their diversity, and the interaction with the host. Direct and indirect defenses promoted by the plant–endophyte symbiont exert an important role in controlling plant defenses against different stresses, and here, more specifically, is discussed the role against biotic stress. Defenses that should be considered are the emission of volatiles or antibiotic compounds, but also the induction of basal defenses and boosting plant immunity by priming defenses. The primed defenses may encompass pathogenesis-related protein genes (PR family), antioxidant enzymes, or changes in the secondary metabolism.

## 1. Introduction

Plant endophytes are described as microorganisms with the capacity to colonize and develop their lifestyle in the inner parts of the plant, including the root, stem, leaf, flowers, and seeds, while not causing apparent damage to the host plant [1]. Normally, endophytes are isolated from surface-sterilized plant tissue and subsequently cultivated in proper medium [2]. However, in recent years, with the development of metagenomics studies, many endophyte communities have been studied through culture-independent approaches such as the sequencing of the 16S rRNA gene and internal transcribed spacer regions (ITS1 and ITS2), the whole genome sequencing, or through shotgun metagenomics studies [3,4,5]. 

Endophytic bacteria play an important role to maintain the health of their host, as they can confer tolerance/resistance to the host plants from abiotic and biotic stresses, as well as in increasing plant growth and crop production [6,7,8,9,10]. Plant–endophyte associations have been studied for many years, however, the mechanisms used by plant endophytic bacteria to mitigate the negative effect of different environmental stressors remains unclear. Several studies have shown that plants may recognize and select their specific microbiota to stablish intimate associations [8,11,12]. Trade-off for such interaction may be based in the capacity of the plant host to provide niches for the microbial partner, and endophytes may produce helpful metabolites and signals [13,14], which can increase nutrient uptake and promote plant growth [15], induce resistance against pathogens [16,17,18,19] and insect herbivores [20], and increase plant tolerance to salinity [21], low temperature [22], heavy metals [23], contaminated chemicals [24], and other abiotic factors. Recent publications have added an additional factor and propose to revisit the disease triangle to consider all the effects produced by the endophytes, in terms of disease suppression from an ecological point of view, going beyond the classic studies [25].

To be able to colonize and move through the plant’s endosphere, endophytic bacteria should be equipped with specific and necessary traits [26]. Several genomic studies showed differences between endophytes, phyllosphere-, and rhizosphere-colonizing bacteria. Different studies have described endophyte-gene candidates involved in the adaptation of the bacteria to the endosphere [21,27] when comparing the complete genomes of different endophytes. Further studies are needed to explain the role of the selected genes for successful colonization, as well as to identify specialized genes conferring the possibility of endophytic lifestyle [28].

The abilities of bacterial endophytes to protect plants against pathogens occur through direct mechanisms, such as the release of antimicrobial compounds such as siderophores, antibiotics, hydrolytic enzymes, and other secondary metabolites, and indirect mechanisms which are related to the competition with pathogens for space and nutrients and their ability to modulate plant defense responses [2,19,29].

Nowadays, most of the research about bacteria endophytes has been done by establishing a parallelism between their action and the plant growth-promoting bacteria (PGP) present in the rhizosphere. However, differences in the rhizosphere or phyllosphere environment from that of internal plant tissues are gaining attention to explain the benefits of endophytes [28]. This review aims to focus on, and to highlight, the impact of the bacterial endosymbionts on the host leading to plant defense by diverse means, but more specifically on the defensive priming phase, a particular immune state of the plant that allows a faster and stronger response to stress once the threat is present, involving different physiological and biochemical changes, helping the host to resist further attacks. Priming defense is a strategy that allows an enhancement of plant defense with low physiological cost [30]. Considering the intimate contact of bacterial endophytes with the plant inner tissues, the onset of the priming phase is not constricted to a specific moment but starts from the colonization process into the plant tissues. Most studies in endophyte-induced resistance against pathogens and pests are performed in terms of description of the phenotype and defenses that are induced after pathogen or pest attack, but it is necessary to consider the defenses that are orchestrated by the presence of endophytic bacteria and those that change only when the pathogen is present. The difficulty to clarify this question arises from the diversity of the microbial community, the plant species, and the known fact that priming is a phenomenon that depends on the specific interaction between the plant and the pathogen. We hypothesize that deciphering the strategies used in priming defenses induced by endophytic bacteria will be the most interesting for future sustainable strategies in crop biotechnology and biocontrol.

The present revision starts with an overview of the diversity and distribution of plant endophytes, then, how they interact with the host to colonize and enter the inner tissues, and finally, the action of the endophytic bacteria in plant defense as a direct impact of the phytopathogens and the indirect actions that may include priming defenses. 

## 2. Diversity and Distribution of Bacterial Endophytes

Bacterial endophytes can be isolated from two different areas known as the phyllosphere (the aboveground part of the plant) and rhizosphere (the belowground part). Each environment harbors different bacterial communities, shaped by the traits of each section. For the belowground, the diversity can be modulated by different factors related to the soil microbial communities and the host factors. The microbiome composition in the root interior is significantly less diverse than the microbiomes in the rhizosphere or in the soil [8]. The number of bacterial cells in the rot internal tissues reaches 10^4^–10^8^ cells per gram of root tissues, while in the rhizosphere, it is around 10^6^–10^9^ bacterial cells per gram [31]. This suggests that roots act as a biological filter, restricting the penetration of mesospheric microbes to the plant endosphere [32]. Regarding the phyllosphere, endophytic bacteria are also less diverse than the epiphytic [33], and it has been rated an average of 10^6^–10^8^ cells bacteria per gram of leaf material [34]. The phyllosphere is in continuous contact with the environment, with an estimated bacterial concentration in the atmosphere of 10^4^ to 10^6^ cells per m^3^ [35], being an important source of bacteria in contact with the plant surface. Additionally, the plant endophyte population also depends on other physiological stages of the plant [11].

Several reports have described the diversity of bacterial endophytes in multiple plant species, particularly those with agronomical interest [36,37]. 

The most representative phylum found in plant bacterial endophytes are Proteobacteria (including α, and γ classes), *Firmicutes*, and *Actinobacteria* [2,3]. Others, such as *Bacteroidetes*, *Planctomycetes*, *Verrucomicrobia*, and *Acidobacteria*, are less commonly reported as endophytes [8]. Among the most representative genera of bacterial endosymbionts, *Pseudomonas*, *Bacillus*, *Burkholderia*, *Stenotrophomonas*, *Micrococcus*, *Pantoea*, *Microbacterium*, *Enterobacter*, *Azospirillum*, and *Serratia* are the most described in the literature [26,32]. Interestingly, although they have been studied as endosymbionts, they are also commonly found in the rhizosphere, supporting the idea that endophytic microbiome may be a subpopulation of the rhizosphere bacteria [8]. Rarefaction analysis has underpinned that the endophytic bacterial diversity found in the phyllosphere is lower than the rhizosphere [38]. Phyllosphere diversity is also drawn by different traits. While the root is already surrounded by the bulk soil and is in direct contact with diverse microbial communities, the aboveground part is exposed to the changing environment and the colonization process may be relayed in the aerial dispersion of the microorganisms or in the landing vectors that can introduce bacteria directly in leaves or stems. The movement of bacterial endophytes along the whole plant endosphere includes a wide spectrum of possibilities, depending on the manner of transmission, the organ, and sub-environments inside the organs [39]. In tomato plants, Dong et al. (2019) [40] showed that the abundance of endophytes was similar in roots and leaves in tomato plants. How this equal distribution in amount is common in different plant species must be elucidated. On the other hand, a recent study presented results pointing to generalist endophytes colonizing the phyllosphere regardless of the plant species, at least in the same geographical area [41], reaching 98% of the detected taxa. Most of the bacteria found belonged to the Proteobacteria and Actinobacteria phylum. Overall, the diversity and localization of endophytic bacteria arise as a complex issue that might be joined to more ecological studies to obtain a more complete picture of the defensive and physiological activities of the hosts.

## 3. Interaction of Bacterial Endophytes with the Host

Plant–endophyte interaction has been developed along millions of years of coevolution, allowing bacteria to develop different traits that enable them to colonize the inner part of the plant [8]. Advanced metagenomic studies point to relevant changes in the genome of bacterial endophytes [3,26]. For instance, a genomic analysis of the endophytic bacterium *Verrucomicrobia* showed a reduction in the genome size with respect to other soil bacteria [40,42]. This reduction in the genome of the obligate endosymbiotic bacteria presents a strategy of adaptation on hosts for several activities [43] and can be explained by the presence of a large expansion of insertion sequences (IS) in endophyte genomes [44].

Endophyte penetration into their host may be mediated by two different processes, either passive or active. In the belowground area, the passive penetration occurs mostly at the site of emergence of lateral roots or in the lesions created by deleterious organisms [11], whereas active penetration may be achieved through the attachment of bacterial endophyte to the plant cells using their flagellum and via production of several metabolites that help in penetration process. Those metabolites include, mostly, exopolysaccharides (EPS), cell wall degrading enzymes, and many other quorum-sensing molecules [26,45]. The presence of bacterial flagellum may mediate endophytic ability allowing bacteria to move by chemotaxis and attach to plant surfaces [46]. Straub et al. (2013) [47] showed that only the endophytic bacteria containing the entire flagella machinery can colonize wheat roots efficiently while flagella-deficient mutant is blocked in colonization abilities. Genes encoding cell wall degrading enzymes, including cellulases, xylanases, cellobiohydrolases, and endoglucanase, were detected in the metagenome of several endophytic bacteria [43,48]. For example, the endoglucanases were shown to be crucial for *Azoarcus* sp. in colonizing rice roots [49], while pectinases secreted by *Bradyrhizobium* sp. are essential for its entrance and translocation in the intracellular space in the same plant species [50]. 

Plant microbiota may be classified ranging from pathogenic to beneficial with different levels of interaction such as epiphytic or saprotrophic [14]. A group of these microorganisms exhibit a mutualistic association with most plants, including the obligate biotrophic lifestyle of some of the microorganisms. Interestingly, they can be found in many varieties of plants including those which have an agricultural interest such as rice, wheat, tomato, maize, strawberry, chickpea, [28,51,52,53], and several wild plants [54,55].

Additionally, several studies suggest that plant genotypes also have a significant influence on the microbiome in the plant endosphere [56,57]. In certain cases, depending on the host plant genotype, the endophytes may have an asymptomatic endophyte lifestyle or pathogenic [58]. For example, *Ramularia collo-cygni* can switch its mode of action from an asymptomatic endophyte to a harmful pathogenic fungus along the developmental stages of crops [59]. In addition, Mina et al. (2020) [33] showed that each genotype and organ of different olive trees are selective towards the phyllosphere endophytic bacteria, the tree genotype being the first selective trait, and then the organ. Thus, the dependence of the host genotype might shape the endophytic microbes as a general characteristic.

### 3.1. Metabolites Implicated in the Interaction of Host–Bacterial Endophyte

Plant colonization of endophytic bacteria involves multiple events, starting with the crosstalk of signal molecules exchanged between the endophytes and their host [26]. Root exudates constitute an important source of metabolites that act as attractants of beneficial microbes. Among the substrates found in the rhizosphere carbohydrates, lipids, amino acids, phenolics, phytosiderophores, and flavonoids are the most described [60] using these molecules as a carbon source for their development. Interestingly, these molecules can modify bacteria–host interaction by modulating gene expression patterns in the microorganism [61,62]. Flavonoids are one of the best described chemoattractants, especially in legume roots with rhizobia [63]. However, these compounds also help non-rhizobial endophytes, such as in the colonization of root in rice by the endophytic bacterium *Serratia* sp. [64]. Other metabolites promoting the attraction of endophytic bacteria are the citric acid and oxalate, from the tricarboxylic acid flux. Specifically, citric acid in cucumber exudates was demonstrated to recruit *Bacillus amyloliquefaciens* SQR9, as well as promote biofilm formation [65]. On the other hand, *Burkholderia phytofirmans* strain defective in oxalate utilization was observed in significantly lower concentration compared to the wild type in both maize and lupine plants secreting different levels of oxalate [66]. Besides, phyllosphere endophytes have easy access to several metabolites, such as carbohydrates and amino acids, among others, but also can access carbon sources derived from plant volatiles such as isoprenes and methanolic compounds that allow microbial populations to develop in the aboveground tissues [51].

### 3.2. Perception of Bacterial Endophytes and Modulation of Plant Immune System

The plant immune system is well defined when describing the plant–pathogen interaction, but when facing bacterial commensalism, the model of immunity is less studied. Bacterial endophytes must circumvent initial plant defenses to colonize the surface of the plant tissue and/or entry into the endosphere. Different approaches have shown that bacterial endophytes, unlike phytopathogens, can emit their microbial-associated molecular patterns (MAMPs) to avoid an overresponse of plant defenses [67]. MAPMs are known to generate different host responses, such as the production of reactive oxygen species (ROS) and phosphorylation cascades, and initiate transcriptional reprogramming and synthesis of secondary metabolites, through the MAMP-triggered immunity (MTI). However, when endophytes enter into communication with the plant, these signals might be different. For instance, Trada et al. (2014) [68] showed that the conserved bacterial MAMP flagellin (flg22) can be differently recognized by the plant from two different strains of bacteria, one pathogenic (*Bacillus phytofirmans*) and the other nonpathogenic (Xanthomonas campestris). It has been discovered that beneficial bacteria *Bacillus subtilis* produces subtilomycin, an antibiotic peptide, that binds its self-produced flagellin, avoiding strong plant response probably because of a non-full perception of the bacteria [69]. Additionally, the protein secretion system (SS) used by bacteria for introducing their effectors in the plant cell is also different. While pathogenic strains use type III and IV SS to deliver their virulent proteins inducing effector-triggered immunity in plants, endophytic bacteria do not use this SS, or if so, in a very low abundance [8]. Another important factor in perception and signaling in plant defense is the production and regulation of ROS. These ROS might be also controlled by some bacteria by producing antioxidant enzymes such as catalases (CAT), superoxide dismutase (SOD), and glutathione-S-transferases (GSTs), among others, at the transcriptional level [3,70]. All these bacterial strategies are framed in the evasion of the plant response by MAMP divergence, by developing variants from the same MAMP, or degradation, by secreting other compounds that can digest somehow their MAPMs [71].

During the establishment of plant–endophyte symbiosis, both actors may modulate the expression of genes related to the colonization and entrance processes [7,8,72]. Interestingly, a comparative genomic study revealed that endosphere isolates of P. fluorescens have significantly more metabolic pathways than those isolated from the rhizosphere that can produce more metabolites used for the plant for signaling events [73]. Recently, several works have evidenced the roles of different miRNA during different pathogenic and mutualistic interactions [74,75,76]. Plants challenged by pathogenic symbionts, most of the miRNAs, appear to act mainly by inducing defensive proteins or targeting detoxification pathways, with the aim of elimination. On the contrary, for the establishment of endophytes, the miRNAs induced in the host during the establishment of symbiotic endophytes target hormone response pathways and innate immune function [74,75,77], reinforcing the plant immunity. A specific example of miRNA during mutualistic interaction includes the miR172c, which promotes nodulation in several plants by suppressing the translation of the ET-inducible transcription factor APETALA2 [78,79]. In general, during the establishment of symbiosis, most pathways targeted by miRNAs for plant defense are turned off, promoting the entrance of the beneficial endophyte [72].

## 4. Extension of Plant Immunity by Endophytes

Benefits derived from the microbiome present in the phyllosphere have been described already in the literature. Recently, two concepts have been defined regarding the modification/amplification of plant immunity due to the microbiome. Considering the holobiome as an entity, the two types of extended immunity that have been proposed are direct and indirect immunity [80]. 

### 4.1. Direct Interactions in the Holobiont

The high diversity of microorganisms in the phytosphere may influence pathogens independently of the plant immune system. Endophytes may adopt several strategies to attenuate the negative impacts of pathogens and pests on their host [2,28]. Those activities may be achieved by direct inhibition of pathogens since they share similar colonizing patterns and are in intimate contact with plants. Events for direct inhibition of pathogens are mainly mediated by inhibitory allelochemicals including siderophores, antibiotics, cell wall degrading enzymes, volatile organic compounds (VOCs), alkaloids, steroids, quinines, terpenoids, phenols, and flavonoids [81,82,83,84,85] (Table 1), or by a quenching signal of pathogens [86]. 

Lipopeptides are one of the most important classes of antimicrobial compounds produced by endophytic bacteria. Lipopeptides isolated from *Bacillus* and *Paenibacillus* are the most studied [87]. Among the *Bacillus* genera, several *Bacillus amyloliquefaciens* strains have been reported as relevant producers of lipopeptides [88]. The endosymbiont *Pseudomonas viridiflava* was reported to also produce antimicrobial compound nominated ecomycins, a lipopeptide containing unusual amino acids including homoserine and β-hydroxy aspartic acid [89]. In addition to lipopeptides, several endophytic isolates were reported to also produce other antibiotic compounds, such as polyketides (surfactin, bacillomycin, fengycin, iturin, lichenysin, mycosubtilin, plipastatin, pumilacidin) produced by *Bacillus subtilis*, polymyxins (a cyclic cationic lipopeptide) synthesized by *Paenibacillus polymyxa* [90], and many other metabolites recently reviewed, such as flavonoids, quinones, alkaloids, phenols, steroids, and terpenoids [91] (Table 1).

Lytic enzymes produced by the endophytes digest a wide variety of polymeric compounds, including chitin, cellulose, proteins, and lipids [8]. One of the actions of plant-colonizing endophytes is the production of enzymes that hydrolyze plant cell walls. These include β-1,3-glucanase, chitinase, cellulase, and protease [92]. Chitinase mediates the degradation of chitin, which is the major cell wall component of fungus, thus the release of these enzymes may exert a cross-action of defense by altering the integrity of fungal cell wall, compromising the survival of the pathogen. For instance, the chitinase produced by endophytic *Streptomyces hygroscopicus* were found to inhibit the growth of different strains of fungi or fungus-like species such as *Ralstonia solani*, *Fusarium oxysporum*, *Alternaria alternata*, *Aspergillus niger*, *Aspergillus flavus*, *Sclerotinia sclerotiorum*, *Hyaloperonospora parasitica*, and *Botrytis cinerea* [93]. The endophytic strain *Bacillus cereus* 65 producing chitinase enzymes was showed to protect the cotton seedlings from root disease caused by *R. solani* [94] (Table 1).

Endophytic bacteria emit VOCs, another group of antimicrobial compounds with a broad-spectrum activity against phytopathogens bacteria, fungi, and nematodes [28,95,96,97,98,99] (Table 1). Sheroan et al. (2016) [96] demonstrated that black pepper-associated endophytic *Pseudomonas putida BP25* could inhibit, by volatile emission, the proliferation of different pathogens including fungi and fungi-like species, and plant-parasitic nematode. In addition to the antimicrobial activity, the advantage of VOCs is their ability to facilitate interactions between physically separated microorganisms. VOCs may be emitted in different chemicals forms. Some of the VOCs produced by endophytes processing as antifungal are cited in Table 1.

Alleviation of ethylene (ET) is also a direct action exerted by endophytic bacteria. ET is demonstrated to increase after pathogen or stress appearance [100], and several reports point to an increase of protection in plants when the seeds have been inoculated with bacterial endophytes. This is due to the bacterial production of the enzyme 1-aminocyclopropane-1-carboxylate (ACC), which can cleave the ET into α-ketobutyrate and ammonia, thus reducing the presence of this enzyme associated to plant stress and physiological damage [101].

An additional mechanism for direct action against pathogens is the quenching of the quorum sensing (QS) required for the survival of most microbes, including pathogens. The QS is responsible for the regulation of physiological activities such as cell–cell crosstalk, reproduction, biofilm formation, adaptation, mutualism, and pathogenesis [102]. Several endophytes were reported to hamper signaling pathways of phytopathogens by quorum-quenching mechanisms [86,103]. For example, the endophytic bacteria harbored in *Cannabis sativa L* was reported to disrupt the cell-to-cell communication of *Chromobacterium violaceum* [103]. Additionally, *Stenotrophomonas maltophilia*, *Pseudomonas aeruginosa*, and *Rhodococcus corynebacterioides* isolated from the xylem of different plant species could degrade the 3-hydroxy palmitic acid methyl ester (3OH-PAME), a QS molecule of *R. solanacearum*, and reduce bacterial wilt in eggplant [104].

### 4.2. Indirect Interactions in the Holobiont

Indirect interactions associated with microbiota comprise the induction of plant defenses. This induction is performed by stimulation of defenses through induced systemic resistance (ISR) or, more accurately proposed in De Kesel et al. 2021 [125], endophyte-induced resistance (E-IR). Each interaction or group of interactions can develop different strategies inducing resistance, depending on the pathosystem.

In general, the priming process depends on different hormonal pathways, [17]. Two types of induced defenses were proposed, ISR and SAR, depending on the hormone implicated and the type of the elicitor [126]. Accordingly, ISR is initiated by rhizobacteria or other non-pathogenic microorganisms, while SAR is induced by pathogens or chemical compounds [127]. The signal transduction pathway of ISR is regulated by the JA/ET pathway and is associated with the expression of the gene *DEFENSIN 1.2* (*PDF1.2*), while SAR is controlled by the SA-dependent signaling pathway and characterized by the expression of genes encoding pathogenesis-related (PR) proteins [128,129,130,131]. More recent studies showed that ISR triggered by endophyte and other rhizobacterial strains may be dependent on SA and dependent or not on JA/ET. Niu et al. (2011) [132] explain the dependency on both SA- and JA/ET-signaling pathways in the ISR mediated by *B. cereus* strain AR156. In addition, the treatment of tobacco roots with *P. fluorescens* CHA0 triggered the accumulation of PR proteins in the leaves induced by SA [133], while in another study, [18], it was shown that the ISR mediated by the root endophytic bacterium *Micromonospora* against *B. cinerea* is dependent only on the JA/ET pathway (Table 2).

The modulation of signaling and defense components during endophytic colonization may result in the activation of the enhanced resistance. Besides, the crosstalk between endophytic communities and the host plant can activate gene clusters, leading to the synthesis of novel secondary metabolites [28,134,135,136]. The ability of endophytes in enhancing plant defenses can be assigned to the interaction of the host plant with the bacterial cell themselves or their metabolites [137]. Numerous studies indicated that ISR can be trigged through several compounds produced by endophytes such as phytohormones, lipopeptides, pyocyanin, siderophore, and VOCs [138,139,140,141]. ISR triggered by endophytic bacteria can protect the host against a wide range of pathogens including soilborne pathogenic fungi [142,143,144], biotrophic pathogens [145,146], viruses [147,148,149], nematodes [150], and insect herbivores [151,152].

The potentiated responses induced by bacterial endophytes include a variety of defense strategies [153]. These strategies include different mechanisms, as illustrated in Figure 1 and discussed below.

#### 4.2.1. Induction of Pathogenesis-Related Proteins (PRs) and Antioxidant Enzymes

Pathogenesis-related proteins (PRs) are well-known for their role in acquired resistance and their induction, triggered by necrotic lesions in plants [154,155]. Different studies evidence the ability of some bacterial endophytic strains to also induce PR activity and eventually increase resistance against a wide range of pathogens [156,157,158]. Among them, β-1,3-glucanases and chitinases (PR-2 and PR-3 families, respectively) are the best described [143]. Benhamou et al. (1998) [159] showed that the endophytic strain *Bacillus pumilus* SE34 triggered ISR in tomato plants, with the elaboration of structural barriers, and production of toxic substances such as phenolic compounds and β-1,3-glucanases. The production of these PR proteins was demonstrated for the disease control exerted by *B. amyloliquefaciens* strain TB2 on litchi downy blight disease caused by *Peronophthora litchi* [156]. In addition, in another study, the endophytic actinobacteria isolated from wheat tissues showed an upregulation of *PR-1*, *PR-4*, *PDF1.2*, and *HEL* defense genes in response to *Erwinia carotovora* subsp. Carotovora challenge [160]. Moreover, recently, endosymbiotic bacteria *Bacillus* spp. was reported to produce antifungal lipopeptides (iturin and fengycin) and induce PR genes, including *PR-1* and *PR-4* in maize [137] (Table 2).

Besides PPRs, several other defense-related enzymes, including antioxidant enzymes SOD, peroxidase (POD), polyphenol oxidase (PPO), phenylalanine, and ammonia-lyase (PAL), were shown to be important in the induced resistance of several endophytes [146,161,162,163]. Chandrasekaran and Chun (2016) [146] showed a significant increase in antioxidant enzymes such as SOD, CAT, PPO, and POD dismutase, catalase, peroxidase, and PPO 24 hours after inoculation of the endophytic bacterium *B. subtilis* in tomato plants. The inoculation of banana plantlet with the endobacterium *S. marcescens* strain UPM39B3 induced the production of host defense enzymes such as peroxidase, PPO, PAL, total soluble phenols, and lignothioglycolic acid, and protected against *Fusarium* wilt disease [164].

#### 4.2.2. Stimulation of Plant Secondary Metabolism

Secondary metabolism of both plant and microsymbiont may be altered due to the interaction. These changes may be produced by several mechanisms. One possibility is that endophyte influences host metabolism by boosting defensive pathways or, in the other way around, plants can manipulate the endophyte metabolism to control its entry and development. Additionally, both plants and bacteria may share different parts of the same pathway, or the host may even metabolize bacterial secretions. In any case, the interplay between host and endophyte is highly controlled by secondary metabolism [165,166].

Phytoalexins are a group of plant antimicrobial molecules with a low molecular weight [131], some of which are terpenoids and flavonoids, among many others. Most studies have been focused on the production of phytoalexins triggered after pathogen perception [167]. On the other hand, some interesting findings revealed that mycorrhizal and rhizobacterial root colonization significantly impact phenolic compounds, alkaloids, terpenoids, and essential oils composition in plants [168,169,170]. Ramos-Solano et al. (2015) [170] demonstrated that level modifications in flavonoids, phenolics, and anthocyanins were associated with delayed postharvest fungal growth on blackberries treated with the *Rhizobacterium* N17.35. Other secondary metabolites, such as the phytoalexin camalexin and glucosinolates, were also quantitatively changed in Arabidopsis plants associated with *Pseudomonas fluorescens* SS101 [171]. In maize, levels of the benzoxaninones were significantly altered in plants associated with rhizobacterial colonization. Further, *Pseudomonas putida* KT2440 inoculated in roots was also shown to modify levels of benzoxaninones after three days after inoculation and cause early responses in maize through JA- and ABA-dependent pathways [172]. 

The abilities of bacterial elicitors such as peptides, glycoprotein, and lipopolysaccharides to trigger plant defense secretion of plant secondary metabolites was reported in several studies. Treatments of the lipopeptide fengycin to potato tuber cells resulted in induced phenylpropanoid pathway metabolism [173]. Moreover, QS molecules of several bacterial groups, such as N-acyl-homoserine lactones (AHLs), were also reported to stimulate an accumulation in phenolic compounds, as well as oxylipins and SA in different plant species [174,175]. 

Endophytes also collaborate with plants in controlling the amount of ROS, thus helping the plants to cope with the toxicity of ROS. This may be explained by the ability of endophytes to produce several metabolites, in particular antioxidants enzymes and phytohormones, as already explained [176,177]. For instance, *Festuca arundinacea*, *Festuca rubra*, and *Elymus dahuricus* contained higher levels of antioxidants and phenolics than endophyte-free plants under different stress conditions [178]. In another study, they found several genes encoding enzymes for ROS scavenging in the genome of several bacterial endophytes, including *Enterobacter* sp. 638 [179], *Gluconacetobacter diazotrophicus* [70], and *Serratia marcescens* RSC-14 [180]. In addition, auxins that could be considered as relevant compounds that mediate plant cell responses to ROS were produced by several endophytes [2,177,181]. Indoleacetic acid (IAA) and other indoles could be isolated from different bacterial endophytes including *Pseudomonas* spp., *Ochrobactrum* spp., *Bacillus* spp., *Arthrobacter*, *Enterobacter*, and *Klebsiella* spp. [182,183,184,185]. Nevertheless, there is scarce knowledge regarding the role of auxins and auxin-related compounds produced by endophytic microbiomes to support plant growth or defense and influence plant–endophyte interactions. 

The endophytic microbiome may also metabolize specific plant-synthesized compounds [186] and may induce the accumulation of other compounds [187]. For example, ET is considered a gaseous hormone that may influence physiological responses to the environment and stress [188,189,190]. Bacterial endophytes could help to enhance the resistance of plants to stress indirectly by decreasing ET levels, especially during stress, when ET concentration increases [8,11]. Some bacterial endophytes use 1-aminocyclopropane-1-carboxylate (ACC), the immediate precursor of ET, as a carbon and nitrogen source by producing ACC deaminase, thereby preventing ethylene signaling [191] and indirectly activating other plant defenses only when the stress is present.

## 5. Conclusions and Perspective

Nowadays, plant immunity is prone to be considered in terms of a whole system, including the action and interaction of a complex holobiome, in which plant and microbes in the phytosphere may prepare the final output when facing biotic stress. In the present study, we have revised the actions exerted by bacterial endophytes in promoting plant health. Moreover, we would like to attract attention to the function of the bacterial endophytes as actors of the ISR and, more specifically, in priming defenses. It is known that primed defenses generate exceptional protection with low physiological cost in the plant. Although the study around this subject is increasing, the mechanisms involved in the joint action of plant–endophyte against biotic stressors are still ongoing, due to the difficulty in working with one specific endophyte strain separately from others sharing the same niche. Several issues are now open in this new field of research. One of these is the choice of studying the action of one endophyte or the combination of several endophytes (which indeed may be closer to the real situation in nature) in induced resistance against biotic stressors; or, perhaps, isolate one strain that can improve plant immunity in different plant species. This can be considered as a three-way interaction study (endophyte–plant–pathogen/pest), gaining more complexity to the experimental system, since it is necessary to know the specific interaction with the plant as the place where the microbes converge. All the works revised in this publication may send a projection of how to develop future strategies for biocontrol, considering the complex interactions between all players, spanning from plants to endophytes, potential vectors of microbes (such as insects), and the whole microbial community that is in contact with the phytosphere. In addition, endophytes may serve as metabolite producers that may help in techniques of biocontrol or for a more sustainable agriculture. Another question that needs to be elucidated is how to combine plant-growth promotion with priming defenses. A possible link between these two events may focus future research.

## Figures and Tables

**Figure 1 plants-10-01012-f001:**
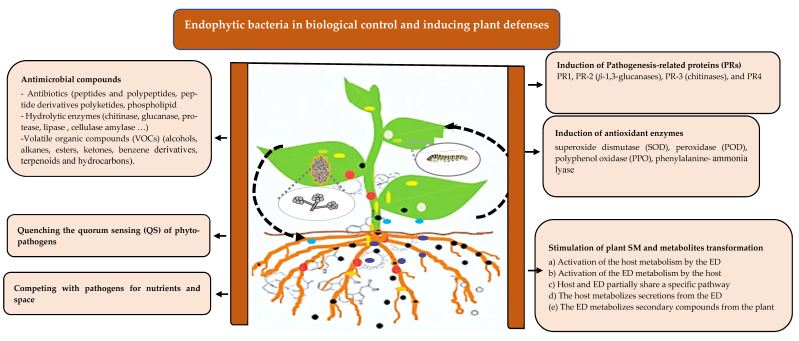
Mechanisms of endophytic bacteria in plant protection against biotic stressors. Endophytic bacteria may protect their host by direct action, producing antimicrobial compounds including antibiotics, hydrolytic enzymes, and VOCs, or indirectly, increasing plant defenses inducing pathogenesis-related proteins, antioxidant enzymes, and stimulation of the secondary metabolisms of both the host and the bacterial endophyte. ED: endophyte; SM: secondary metabolism.

**Table 1 plants-10-01012-t001:** Antimicrobial compounds produced by the bacterial endophytes.

	Endophytes	Plant Host Class	Activity	Compounds	Chemical Classes	References
**Antibiotics**	*Pseudomonas viridiflava*	Grass	Antifungal	Ecomycin	Peptide	[89]
	*B. subtilis* 168		Antifungal	Bacilysocin	Phospholipid	[105]
	*B. thuringiensis*		Insecticidal	β-exotoxin	Polypeptide	[106]
	*Streptomyces* sp. strain NRRL 30562	*Kennedia nigriscans*	Antifungal/Antibacterial	Munumbicins A, B, C, and D,	Peptide	[107]
	*Streptomyces* sp. strain NRRL 30566	*Grevillea pteridifolia*	Antibacterial	Kakadumycin A	Peptide	[108]
	*Verrucosispora maris* AB-18-032	*Sonchus oleraceus*	Antibacterial	Proximicin	Peptide	[109]
	*Streptomyces* sp. HK 10595	Kandelia candel	Antibacterial	Xiamycin B, Indosespine and Sespenine	Entacyclic indolosesquiterpine	[110]
	*Streptomyces* sp.	marine mudflat-derived actinomycete	Antibacterial	Harmaomycin	Peptide derivatives	[111]
	*B. substilis*		Antibacterial	Subtilin	Peptides	[112]
	*Bacillus atrophaeus*, *Bacillus mojavensis*	Glycyrrhiza uralensis	Antifungal	1,2-bezenedicarboxyl acid, Methyl ester, Decanodioic acid, bis(2-ehtylhexyl) ester	Polyketides	[113]
	*Lysinibacillus*, *Staphylococcus*, *Enterobacter*, *Pseudomonas*, and *Bacillus* species	Combretum molle	Antibacterial			[114]
	*Bacillus* *subtilis* strain 1-L-29	*Camellia oleifera*	Antifungal			[115]
**VOC_S_**	*Nodulisporium* sp.	*Myroxylon balsamum*	Antifungal	phenylethyl alcohol. alkyl alcohols alkyl alcohols (1-butanol-3-methyl, 1-propanol-2-methyl, 1-pentanol, 1-hexanol, 1-heptanol, 1-octanol, 1-nonanol	esters, ketones, benzene derivatives, a terpenoids, hydrocarbons.	[116]
	*Enterobacter aerogenes*	*Maize*	Antifungal	2,3-butanediol		[117]
	*Pseudomonas putida* BP25		Antifungal/antiparasitic			[96]
	*Bacillus velezensis* ZSY-1,	Chinese catalpa	Antifungal	2-tridecanone, pyrazine (2,5-dimethyl), benzothiazole, and phenol (4-chloro-3-methyl)	ketones, alcohols, and alkanes	[118]
	*Bacillus* spp.		Antifungal	1-Hexadecanol, Hexacosyl acetate, Tryphenylphosphine oxide, 1,3-Propanediol, 2-methyl, dipropanoate, 1,4-Pentadiene, Hydroxyurea, Decyl trifluoroacetate, Pentadecane, 4-Ethyl-1-octyn-3-ol, Tridecane Benzothiazole, *N*,*N*-Dimethyldodecylamine, 1,3-Butadiene, Dodecane, 2-Undecanone IR-(+)-a-pinene		[119]
	*Bacillus subtilis* DZSY21	*Eucommia ulmoides*	Antifungal	2-Methylbutyric acid, 2-heptanone, and isopentyl acetate		[120]
**Enzymes**	*B. cereus* 65	mustard	Antifungal	chitinase		[93]
	*Lysobacter enzymogenes*	Sugar Beet	Antifungal	1,3-glucanase		[121]
	*Streptomyces hygroscopicus*			chitinase		[94]
	*Bacillus pumilus* JK-SX001	*Populus*	Antifungal	cellulases and protease		[122]
	*Bacillus* sp., *Micrococcus* sp., and *P. polymyxa*	*Panax ginseng-* and *Plectranthus tenuiflorus*	Antibacterial	amylase, esterase, lipase, protease, pectinase, xylanase, and cellulase		[123]
	*Paenibacillus* sp. and *Bacillus* sp.	*Lonicera japonica*		cellulase and pectinase		[124]

**Table 2 plants-10-01012-t002:** Different mechanisms of defenses induced by endophytic bacteria.

Endophyte	Host Plant	Application of Endophytes	Plant Pathogens	Signaling Pathway	Induced Defenses	Reference
*B. pumilus* SE34	Tomato	Germinated seeds	*Fusarium oxysporum* f. sp. *radicis lycopersici*		Elaboration of structural barriers, production phenolic compounds and β-1,3-glucanases.	[159]
*Serratia plymuthica*	Cucumber	Seeds	*Pythium ultimum*		Callose-enriched wall appositions at sites of pathogen penetration. Accumulation of an osmiophilic material in the colonized areas.	[142]
*B. mycoides*	Sugar beet	Foliar application	*Cercosporonbeticola Sacc.*		Increased activity of B-1,3-glucunase, chitinase, peroxidase and PR proteins.	[192]
*Bacillus subtilis* GB03 and *Bacillus amyloliquefaciens* IN937	*A. thaliana*	Exposition to the VOCs produced by the isolates	*Erwinia carotovora* subsp. carotovora	ET pathway		[139]
Actinobacteria	*A. thaliana*	Roots	*E. carotovora sub*sp. *Carotovora*	JA/ET pathway	Upregulation of, *PR-1*, *PR-4* as well as *PDF1.2* and *HEL* genes.	[160]
*B. amyloliquefaciens strain TB2*	Lychee	Fruit	*Peronophthora litchi*		Increase PPRs production including 1,3-glucanase and chitinase.	[156]
*Bacillus cereus AR156*	Tomato	Roots	*Pseudomonas syringae pv. tomato*	Simultaneous activation of SA- and JA/Et-dependent signaling pathways	Activation of some defense-related genes including *PR1*, *PR2*, *PR5*, and *PDF1.2.*	[132]
*Bacillus B014*	Anthurium	Foliar application	*Xanthomonas axonopodis pv. dieffenbachiae*		Activation of defense-related enzymes PAL, POD and PPO after pathogen attack.	[193]
*Enterobacter radicincitans DSM 16656*	*A. thaliana*	Roots		Both SA- and JA/Et-dependent signaling pathways	Accumulation of *PR1*, *PR2*, *PR5*, and *PDF1.2* transcript 24 h after treatment with the endophyte.	[194]
*Bacillus* spp.	Maize	Roots			Upregulation of pathogenesis-related genes.	[137]
*B. amyloliquefaciens S499*	Tomato	Seeds and soil	*Botrytis cinerea*		Induction of lipoxygenase pathway (accumulation of transcripts of genes corresponding to the two isoforms, *Tom LOXD* and *Tom LOXF.*	[195]
*Pseudomonas fluorescens PICF7*	Olive	Roots	*Verticillium* wilt		Activation of olive genes potentially coding for lipoxygenase 2, catalase, 1-aminocyclopropane-1-carboxylate oxidase, and phenylananine ammonia-lyase.	[196]
*B. amyloliquefaciens* strain FZB42	Lettuce	Roots	*R. solani*		Higher expression of *PDF* 1 and 2.	[197]
*B. amyloliquefaciens* strain Blu-v2	Hosta	Filiar application	*Spodoptera fruigiperda*		Production lipopeptides that elicits ISR in plants against fall armyworms.	[151]
*Micromonospora* spp.	Tomato	Roots	*B. cinerea*	JA/ET pathway	Induction of JA-regulated defenses (*PINII* and *LOX* A).	[18]
*P. fluorescens WCS417r*	*A.thaliana*	Soil drench	Chewing herbivores	JA/ET	Increased expression of *LOX2*, *PDF1.2* and *HEL.*	[198]
*P. simiae* WCS417r	*A. thaliana*	Roots	Leaf-chewing herbivores	JA/ET	Higher expression of *ORA59*-branch respect to the JA-dependent *MYC2* branch, Enhanced synthesis of camalexin and aliphatic glucosinolates (GLS).	[199]
*Bacillus* sp.	Cotton	Soil drench	Beet armyworm *Spodoptera exigua*	JA	Accumulation of JA, and JA-related genes.	[200]
*B. pumilus*	Grapevine	Soil drench	*P. chlamydospora*		Enhanced expression of different genes PR 1, PR 10, chitinase class III, PAL, stilbene synthase, chalcone synthase, anthranilate synthase, callose synthase, Glutathione S-transferase, and b-1,3 glucanase.	[201]
*B. subtilis*	Tomato	Soil drench	*Xanthomonas campestris* pv. *vesicatoria*		Accumulation of antioxidant enzymes SOD, CAT, POD, and PPO.	[146]
*Bacillus amyloliquefaciens strain MBI600*	Tomato	Drench or foliar application	*Tomato spotted wilt virus* and *Potato virus Y*	SA pathway	Induction of the SA signaling pathway in tomato after MBI600 treatment.	[148]
*Azospirillum* sp. B510	Rice	Soil drench		ET signaling is required for endophyte-mediated ISR	Induces systemic disease resistance in rice without accompanying defense-related gene expression.	[76]
*Streptomyces* spp.	Rice, sorghum and wheat	Seedlings			Upregulation of *PR10*, *NPR1*, *PAL* and *LOX2.*	[202]
*Bacillus cereus* *Serratia nematodiphila TLE1.1*	Tomato	Seeds	*Ralstonia syzigiisub* sp.	JA	Increase JA contained in leaves and roots of tomato significantly until 12 days after pathogen inoculation.	[203]

## Data Availability

Not applicable.

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
