# Peer review of "Bacterial Endophytes: The Hidden Actor in Plant Immune Responses against Biotic Stress"

_plants, 2021, doi:10.3390/plants10051012_

Round 1

Reviewer 1 Report

The manuscript by Oukala et al. reviews the role of endophytic bacteria in modulation of plant immune responses to biotic stress. The review is comprehensive and well organised. It covers the topic in a good level of detail. Of particular value is the emphasis on the mechanisms of beneficial action specific to endophytes that are distinct from those of external beneficial microbes such as plant growth promotion rhizosphere bacteria. The review goes into excellent detail of the two main areas by which endospheric bacteria modulate the plant immune system.

The review also includes a very helpful tabular summary of the antimicrobial compounds produced by bacterial endophytes and main plant defences induced by endophytic bacteria along with their mechanisms of induction.

Overall, this is a very useful review of the field which will be of interest to a broad readership within the plant sciences.

The review could possibly add more about phyllospheric endophytes. Currently, the focus is on the root system, particularly with regard to entry to the plant but plants also have an extensive phyllospheric endosphere bacterial population. It would be good to see some brief consideration of the origin of this population too and reflection of whether this population is equivalent to the root population in terms of plant immune system modulation.

Minor points

Where authors of papers are referred to as part of the text, the authors’ names should be included as well as just the numeric reference code. For example line 48, “identified by [20,25]” should be “identified by Subramanian et al. [20] and Khare et al. [25]”.

Line 75: “root” not “rot”.

Line 82: The class Betaproteobacteria has recently been reclassified as the new order Betaproteobacteriales within the class Gammaproteobacteria. (​Parks et al., 2018.. Nat. Biotechnol. 36:996–1004).

Line 169, 170, 171, 318, 319, 322, possibly others: Latin binomials should be in italics.

Line 178: “bacteria” not bacterias”. Bacteria is already plural (singular bacterium).

Line 242: B. cereus should be written in full at the first mention here. (In fact, Bacillus cereus is written in full later in the manuscript).

Line 318: B. amyloliquefaciens should be written in full at the first mention here.

Line 369: plants under stress conditions had more antioxidants than endophyte-free plants. Presumably, both sets of plants are under stress conditions and the plants more antioxidants had endophytes.

Figure 1 defines most abbreviations used in the figure but could also add definitions for SM and ED.

There are also minor grammatical errors throughout.

Reviewer 2 Report

Review Comments

This review reviewed the role of endophytic bacteria in modulating the defence of the host rendering the entire plant more resistant to pathogens and pests. The endophyte-induced resistance may probably introduce a new factor when considering plant-pathogen interactions. The impact of the bacterial endosymbionts on the host leading to the priming state is also discussed, however some improvements and revisions are still required as shown below;

- The abstract is badly written. Please improve the abstract to cover the important topics reviewed and discussed in this article.

- The introduction should be improved by highlighting the published studies related to this topic. Furthermore, the hypothesis and objectives of this review should be mentioned clearly in the introduction section.

- I may suggest adding a section on “the plant growth-promoting traits of bacterial endophytes. This new section will help illustrating the mechanisms behind that.

- Additionally, presenting a figure for these mechanisms (plant growth-promoting traits) will be interesting.

- Section 2 should also discuss some enough literature on the recent related works that could be interpreted with it.

- The English and language grammar should be revised and corrected throughout the whole manuscript

- Figures should be clearly interpreted with the authors’ discussion.

- The conclusion section should mention the importance and future perspectives as well.

- Bibliography section should be revised and updated as per the suggested literature mentioned above.

Reviewer 3 Report

Excellent review on Biologicals - the most important subject in global agriculture.  Very comprehensive and detailed on bacteria species, modes of action and physiological impact on crop production. This review also impacts global plant health, critical for sustainable crop production. Literature review very comprehensive.

Reviewer 4 Report

Comments on plants-1192921

The authors have focused on summarizing the latest information available on the role of bacterial endophytes in alleviating biotic stresses which is quite interesting

While discussing the role of endophytes under stress, their indirect effect in the promotion of growth needs more elaboration. The following paper may be considered: https://doi.org/10.3390/ijerph17238859

The authors need to give a little more focus on the alleviation of ethylene, which is both produced under biotic and abiotic stress. In addition, materials should be added on the production of ethylene during biotic stress, its concentration and the role of endophytes in reducing these concentrations, preferably a Table may be added  

There are many typos, which need thorough crosscheck, e.g. the title of Table 2 and many others like that. All the Tables and even Figures need correction regarding these types of mistakes

In Table 1, what do you mean by “Grasse”

Another subsection may be added on the role of endophytes in biotic stress alleviation through exogenous application via foliar, seed, or root primed

More rigorous discussion should be added in each section with a special focus on the research gaps

Why this review is important or what it contributes to the previous knowledge need to be added at the end of the Introduction section

The abstract section is very short, major findings of this review may be added

Round 2

Reviewer 2 Report

The revised manuscript has been greatly improved as per the suggested comments raised. So, I could recommend it for publication now

Reviewer 4 Report

The paper has been improved very much and can be considered for publication